# Nuclear Fragility in Radiation-Induced Senescence: Blebs and Tubes Visualized by 3D Electron Microscopy

**DOI:** 10.3390/cells11020273

**Published:** 2022-01-13

**Authors:** Benjamin M. Freyter, Mutaz A. Abd Al-razaq, Anna Isermann, Anne Dietz, Omid Azimzadeh, Liesbeth Hekking, Maria Gomolka, Claudia E. Rübe

**Affiliations:** 1Department of Radiation Oncology, Saarland University Medical Center, Kirrbergerstrasse Building 6.5, 66421 Homburg, Germany; benjamin.freyter@uks.eu (B.M.F.); mutaz.abd-al-razaq@uks.eu (M.A.A.A.-r.); anna.isermann@uks.eu (A.I.); 2Department of Effects and Risks of Ionising & Non-Ionising Radiation, Federal Office for Radiation Protection, 85764 Oberschleißheim, Germany; adietz@bfs.de (A.D.); oazimzadeh@bfs.de (O.A.); MGomolka@bfs.de (M.G.); 3Thermo Fisher Scientific, 5651 Eindhoven, The Netherlands; liesbeth.hekking@thermofisher.com

**Keywords:** cellular senescence, ionizing radiation, radiation-induced senescence, chromatin reorganization, nuclear blebbing, transmission electron microscopy (TEM), serial block-face scanning electron microscopy (SBF-SEM), cytosolic chromatin fragments (CCF), cGAS-STING signaling

## Abstract

Irreparable DNA damage following ionizing radiation (IR) triggers prolonged DNA damage response and induces premature senescence. Cellular senescence is a permanent state of cell-cycle arrest characterized by chromatin restructuring, altered nuclear morphology and acquisition of secretory phenotype, which contributes to senescence-related inflammation. However, the mechanistic connections for radiation-induced DNA damage that trigger these senescence-associated hallmarks are poorly understood. In our in vitro model of radiation-induced senescence, mass spectrometry-based proteomics was combined with high-resolution imaging techniques to investigate the interrelations between altered chromatin compaction, nuclear envelope destabilization and nucleo-cytoplasmic chromatin blebbing. Our findings confirm the general pathophysiology of the senescence-response, with disruption of nuclear lamin organization leading to extensive chromatin restructuring and destabilization of the nuclear membrane with release of chromatin fragments into the cytosol, thereby activating cGAS-STING-dependent interferon signaling. By serial block-face scanning electron microscopy (SBF-SEM) whole-cell datasets were acquired to investigate the morphological organization of senescent fibroblasts. High-resolution 3-dimensional (3D) reconstruction of the complex nuclear shape allows us to precisely visualize the segregation of nuclear blebs from the main nucleus and their fusion with lysosomes. By multi-view 3D electron microscopy, we identified nanotubular channels formed in lamin-perturbed nuclei of senescent fibroblasts; the potential role of these nucleo-cytoplasmic nanotubes for expulsion of damaged chromatin has to be examined.

## 1. Introduction

Cellular senescence is a complex stress response leading to permanent cell cycle arrest with resistance to mitogenic or apoptotic signals [1]. Senescent cells are characterized by extensive chromatin restructuring with formation of senescence-associated heterochromatic foci (SAHF) and altered nuclear morphology. Senescent cells acquire the senescence-associated secretory phenotype (SASP), with increased production of pro-inflammatory cytokines [2,3]. Accumulation of senescent cells during aging has been implicated in promoting various age-related diseases [4,5]. As cellular senescence is characterized as the basis for aging in organisms, culturing of normal human fibroblasts to mimic the in vivo aging processes has been developed as major methods to investigate cellular and molecular events involved in aging [6]. 

Normal cells undergo premature senescence in response to severe or irreparable DNA damage, induced, e.g., by ionizing radiation (IR) [3]. Radiation-induced DNA damage, such as double-strand breaks (DSBs), triggers DNA damage response (DDR) mechanisms to coordinate DSB repair activities within the chromatin context [7,8]. During this DDR, multiple repair proteins are recruited to DSB sites, forming radiation-induced foci (e.g., γH2AX- and 53BP1-foci) [9,10]. Focal accumulation of DSB repair factors around persisting DNA lesions is a characteristic feature of cellular senescence [11]. 

Senescent cells undergo striking morphological changes with irregular-shaped nuclei characterized by protrusions and invaginations. Recent work highlighted the important role of the nuclear envelope in regulating chromatin organization and maintaining nuclear stability [12]. The nuclear envelope consists of an outer nuclear membrane continuous with the endoplasmic reticulum, and an inner nuclear membrane facing the nucleoplasm and nuclear pore complexes. The inner surface of the nuclear envelope is lined by lamina, which contributes to the size, shape, and stability of the nucleus [13]. Major structural components of this filament meshwork are the nuclear lamins, categorized as A-type and B-type lamins. Opening of these separate but interacting lamin meshworks is responsible for nuclear bleb formation [14]. Lamins have also been found to interact with chromatin by forming physical connections at the nuclear periphery [15]. These structural associations of the nuclear lamina with peripheral chromatin called lamina-associated domains (LADs) mediate functional organization of the genome [16,17]. Previous studies in oncogene-induced senescence have shown that the dissociation of the lamina from peripheral heterochromatin is associated with the formation of SAHFs [18].

Persistent DDR signaling is essential to establish and maintain the SASP in senescent cells [3,19]. However, the mechanisms linking DDR and SASP in radiation-induced senescence are incompletely explored. Recent studies suggest that nucleus-to-cytoplasm blebbing of chromatin with the formation of cytoplasmic chromatin fragments (CCFs) may connect these central concepts in cellular senescence [20,21]. Chromatin fragments released into the cytosol by nuclear envelope ruptures can be recognized by cyclic GMP-AMP synthase (cGAS) protein, and may thereby activate stimulator of interferon genes (STING). This cGAS-STING signaling pathway is a first line defence component of the innate immune system that mediates type I interferon inflammatory responses to repel viral and bacterial infections [22]. Only recently has it been shown that the cGAS-STING signaling is also activated by endogenous DNA aberrantly localized in the cytosol, such as extranuclear chromatin resulting from genotoxic stress [23]. Accordingly, recognition of CCFs by the cGAS-STING pathway could be essential for the induction of SASP in senescent cells [22].

During senescence progression, the precise mechanisms governing the assembly and expulsion of damaged chromatin through the nuclear membrane are still unclear. Combining proteome analysis with different high-resolution imaging techniques, we comprehensively investigate the interrelations between altered chromatin compaction, nuclear envelope destabilization, and nucleus-to-cytoplasm blebbing during radiation-induced senescence.

## 2. Materials and Methods

Cell culture: Human WI-38 fibroblasts were obtained from the American Type Culture Collection (ATCC NPO, Manassas, VA, USA). Fibroblasts were cultured at 5% O_2_ and 5% CO_2_ in Minimum Essential Media (MEM; Invitrogen, Karlsruhe, Germany) with 10% fetal bovine serum, 1 mM sodium pyruvate, 2  mM L-glutamine, 0.1 mM non-essential amino acids, and 1% penicillin/streptomycin. For monolayer experiments, cells were grown on coverslips to 90% confluence before starting experiments; for protein analysis cells were grown in T75 flasks.

Radiation exposure: Fibroblasts were irradiated with 20 Gy using Artiste™ linear accelerator (Siemens, Munich, Germany) (6 MV photons; dose-rate 2 Gy/min). After IR exposure fibroblasts were cultured for 2 weeks and compared to non-irradiated controls.

Cytochemical detection of SA-β-Gal: Following 5 min fixation with 2% paraformaldehyde and 0.2% glutaraldehyde, cells were incubated with X-Gal staining solution (AppliChem GmbH, Darmstadt, Germany) at 37 °C overnight. After 30 s methanol incubation, dried samples were permeabilized with 0.2% TritonX-100 and washed with 1% BSA. Samples were blocked with 4% BSA for 1 h, followed by an overnight incubation with anti-H2A.J primary antibody. Incubation with Dako biotinylated secondary antibody (Agilent, Waldbronn, Germany) was followed by Vectastain ABC Peroxidase (Vector, Burlingame, CA, USA) and SIGMAFAST™ 3,3′diaminobenzidine (Merck, Darmstadt, Germany) incubations, respectively. Samples were mounted in Dako Faramount Mounting Medium (Agilent, Waldbronn, Germany).

2′3′-cGAMP ELISA: 2 weeks after (sham-) IR exposure, medium of 20 Gy-irradiated and non-irradiated fibroblasts was aspirated and cells were gently washed with phosphate-buffered solution (PBS; ThermoFisher Scientific) and lysed using M-PER™ Mammalian Protein Extraction Reagent. Lysates were centrifuged at 13,000× *g* for 10 min at 4 °C. 2′3′-cGAMP concentration was measured in supernatants using 2′3′-cGAMP ELISA assay (Cayman Chemical, Ann Arbor, MI, USA), and data were analyzed using four-parameter log fit according to manufacturer’s manual.

Immunofluorescence microscopy (IFM): Cells were fixed with 4% paraformaldehyde and permeabilised with 0.5% TritonX-100, washed with 0.1% Tween^®^-20 and incubated overnight with primary antibody (anti-p21, Abcam, Berlin, Germany; anti-Ki67 ThermoFisher, Waltham, MA, USA; anti-BrdU, Bio-Rad Laboratories, Munich, Germany; anti-Lamin B1, Proteintech, Manchester, UK; anti-H3K27me3 and anti-H3K9me3, Abcam, Berlin, Germany; anti-53BP1, Bethyl, Montgomery, TX, USA; anti-γH2AX, MilliporeSigma, Sigma-Aldrich Chemie GmbH, Germany) followed by Alexa-Fluor^®^488 or Alexa-Fluor^®^568 secondary antibody (Invitrogen, Karlsruhe, Germany). Subsequently, cells were mounted in VECTAshield™ mounting medium with 4′,6-diamidino-2-phenylindole (DAPI, Vector Laboratories, Burlingame, CA, USA). Fluorescence images were captured with Nikon-Eclipse Ni fluorescence microscope equipped with Nikon DS-Qi2 camera (Nikon, Düsseldorf, Germany). For evaluating SA-β-Gal, p21, Ki67 and BrdU positivity, at least 200 cells were captured for each sample (positive cells in %). For the measurement of lamin B1 fluorescence intensity the FITC signal was normalized to DAPI signal in 200 cells using Nikon NIS-Elements Basic Research acquisition software (Nikon, Düsseldorf, Germany). 

BrdU labeling: To test the stability of senescence-associated growth arrest, 20 Gy-irradiated and non-irradiated cells were pulsed with 10 µmol/L 5-bromo-2′-deoxyuridine (BrdU) in culture medium for 24 h. After medium removal and PBS washing steps, cells were fixed and permeabilized as described for standard IFM. DNA denaturation was completed through 1 h incubation in 2 M HCl. Following, the cells were washed with PBS and incubated for 2 h in primary anti-BrdU antibody (Bio-Rad Laboratories, Munich, Germany) in 0.1% Tween, 1% bovine serum albumin (BSA) in PBS. After washes of Tween/BSA/PBS, samples were incubated with fluorescence-coupled anti-rat secondary antibody in Tween/BSA/PBS for 2 h. After PBS washes, coverslips were mounted with hard-set mounting medium containing DAPI.

Protein extraction and determination of protein concentration: Cell pellets (1 × 10^6^) were thawed on ice before lysed in radio-immunoprecipitation assay (RIPA) buffer containing 25 mM Tris.HCl (pH 7.6), 150 mM NaCl, 1% NP-40, 1% sodium deoxycholate, 0.1% SDS + 1× Halt™ Protease and Phosphatase Inhibitor Cocktail (Thermo Fisher Scientific, USA). The reaction was suspended and incubated on ice for 15 min followed by 5 min incubation at 95 °C. To achieve efficient cell disruption, reaction was subjected to Bioruptor^®^ Pico sonication device (Diagenode, Belgium) with 5 sonication cycles (30 s ON/30 s OFF) at 4 °C followed by centrifugation (10 min, 22,000× *g*, 4 °C). Supernatants containing protein lysate were transferred to LoBind tubes. Protein concentrations were determined in duplicate by RC DC™ kit as recommended by the manufacturer (Bio-Rad Laboratories, Inc., Hercules, CA, USA) using bovine serum albumin (BSA) as standard on Infinite M200 Spectrophotometer (Tecan GmbH, Germany). For mass spectrometry analyses, 5 μg of each protein lysate was used. 

High-performance liquid chromatography online coupled to tandem mass spectrometry (HPLC-MS/MS): Protein lysates were digested using modified filter-aided sample preparation (FASP) protocol, as described [24,25]. MS data were acquired on Q-Exactive HF-X mass spectrometer (Thermo Fisher Scientific, Waltham, MA, USA) coupled to nano-RSLC (Ultimate 3000 RSLC; Dionex, Thermo Fisher Scientific, Waltham, MA, USA) [26]. Proteome Discoverer (PD) 2.4 software (Thermo Fisher Scientific; version 2.4.1.15) was used for peptide and protein identification via database search (Sequest HT search engine) against the Swiss-Prot human database (Release 2020_02, 20349 sequences in PD). Peptide spectrum matches and peptides were performed by accepting only top-scoring hit for each spectrum, and satisfying cut-off values for FDR < 1%, and posterior error probability < 0.01. The final protein ratio was calculated using the median abundance values of 4 replicates for each of the experimental groups. Statistical significance of the ratio change was ascertained employing the *t*-test approach. For final quantifications, proteins identified with more than one unique peptide in at least 2 of 3 replicates and having ratios greater than 1.5-fold or less than 0.66-fold (adj. *p*-value < 0.05) were defined as being significantly differentially expressed. Pathway analysis was performed using ReactomePA.

Transmission electron microscopy (TEM): Cells were fixed overnight using 2% paraformaldehyde and 0.05% glutaraldehyde in PBS. Samples were dehydrated in increasing ethanol concentrations and infiltrated overnight with LR White resin (EMS, Hatfield, PA, USA) followed by overnight embedding at 65 °C with fresh LR White resin containing LR White Accelerator (EMS, Hatfield, PA, USA). Microtome Ultracut UCT (Leica, Wetzlar, Germany) and diamond knife (Diatome, Biel, Switzerland) were used to acquire ultrathin sections (70 nm) picked up on pioloform-coated nickel grids and prepared for immunogold-labelling. Non-specific labeling was blocked using Aurion blocking solution (Aurion, Wageningen, The Netherlands), sections were rinsed and incubated overnight with primary antibodies (anti-Lamin B1, Proteintech, Manchester, UK) at 4 °C, followed by incubation with 10 nm gold particle-conjugated secondary antibodies (Aurion, Wageningen, The Netherlands) for 1.5 h. Finally, sections were contrasted with uranyl acetate. Tecnai Biotwin™ transmission electron microscope (FEI, Eindhoven, The Netherlands) was employed for visual analysis.

Serial block-face scanning electron microscopy (SBF-SEM): Cells were fixed using 1.25% glutaraldehyde, 2.5% paraformaldehyde, and 2 mM CaCl_2_ in 0.15 M cacodylate buffer (pH 7.4; EMS, Hatfield, PA, USA) at 4 °C, followed by 2% OsO_4_, 1% uranyl acetate and en-block lead aspartate contrasting steps and ethanol/acetone step-wise dehydration. Samples were infiltrated overnight using increasing proportions of Durcupan (EMS, Hatfield, PA, USA) in acetone solutions. Embedding steps were performed at 65 °C with fresh Durcupan resin. Finally, samples were detached from coverslips by shrinking Durcupan resin by rapid temperature reduction and attached to sample holders for SEM imaging. For high-resolution imaging of the ultrastructure of senescent fibroblasts, serial block-face imaging was performed in VolumeScope™ SEM (Thermo Fisher Scientific, Waltham, MA, USA). Serial block-face imaging combines mechanical sectioning using a chamber microtome with automated image acquisition to obtain large sample volumes. Slice thickness was 50 nm and images were acquired in low vacuum (30 Pa) with 2 kV, current of 100 pA and dwelltime of 2 µs. For 3D reconstruction, the entire volume of whole cells was captured through stacking of sequential images and micrographs were manually segmented using analytical software Amira 6.7.0 (Thermo Fisher Scientific, Waltham, MA, USA). 

Statistical Analysis: Data were presented as mean ±SEM, where normally distributed data were analyzed by Student’s *t*-test to evaluate differences between 20 Gy-irradiated and non-irradiated cells. Statistical analyses were performed by Graphpad Prism 9.2.0 (Graphpad Software, San Diego, CA, USA). Statistical significance was presented as * *p* < 0.05, ** *p* < 0.01, *** *p* < 0.001.

## 3. Results

### 3.1. Premature Senescence after High-Dose IR Exposure

Previous studies in our lab showed that irreparable DSBs in response to IR trigger prolonged DDR and induce premature senescence in human WI-38 fibroblasts [3]. 

After IR exposure (20 Gy, 2 w post-IR), ≈90% of WI-38 fibroblasts showed increasing staining intensities for the senescence marker SA-β-Gal and p21, suggesting progressive entry of these cells into radiation-induced senescence (Figure 1A). Reduced proliferation rates of 20 Gy-irradiated fibroblasts were verified by clearly diminished Ki67 expression and reduced BrdU incorporation (Figure 1B). Moreover, the expression of lamin B1 was explored by IFM and TEM. In non-irradiated fibroblasts lamin B1 was evenly expressed at the inner surface of the nuclear membrane, but in 20 Gy-irradiated fibroblasts lamin B1 was strongly down-regulated (Figure 1C). For lamin B1 the relative fluorescence intensity (IFM: FITC signal normalized to DAPI by Nikon NIS-Elements™) and relative protein abundance (proteome analysis by HPLC-MS/MS; LMNB1 in Appendix A) was measured in non-irradiated versus irradiated fibroblasts. Both measurement methods for lamin B1 quantification confirmed the senescence-associated lamina decline, potentially affecting the integrity and stability of the nuclear membrane (Figure 1C). Our data indicate that radiation exposure to 20 Gy induces reliably premature senescence in human fibroblasts and thus represents an excellent model to study the nuclear fragility during radiation-induced senescence.

### 3.2. Proteome Analysis 

Label-free proteome analysis was performed to generate comprehensive protein profiles in 20 Gy-irradiated versus non-irradiated WI-38 fibroblasts. Principal component analysis (PCA) (based on normalized intensity of all identified proteins) indicates that 20 Gy- and non-irradiated samples clustered into two separate groups (PC1 49.7% and PC2 24.7%) (Appendix A). Among 4142 quantified proteins with at least two unique peptides (Appendix A), the expression of 96 proteins was significantly different (+1.5-fold; adj. *p*-value < 0.05) after IR. Out of those, 58 proteins were down- and 38 up-regulated (Figure 2A and Appendix A). Detailed analysis of functional interactions was performed using R package ReactomePA to screen for different pathways involved in radiation-induced senescence. Most affected pathways in 20 Gy-irradiated versus non-irradiated fibroblasts were related to DNA damage/telomere stress induced senescence, chromatin organization and SAHF formation (Figure 2B and Appendix A). Accordingly, our proteome data are in line with previous studies, showing that cellular senescence is characterized by extensive chromatin restructuring with global histone H1 loss and SAHF formation, as well as with nuclear degradation of lamin B1 (Appendix A). Moreover, our proteome data suggest that interferon signaling may drive the production of inflammatory SASP components (Figure 2C). 

### 3.3. Chromatin Remodeling and Nuclear Blebbing during Radiation-Induced Senescence Visualized by IFM

Following IR exposure, human WI-38 fibroblasts undergo striking morphological changes during senescence progression, which can be visualized by IFM. While non-irradiated fibroblasts harbor more or less elongated, but smoothly ovoidal nuclei, senescent fibroblasts following IR exposure reveal enlarged nuclei with distorted morphologies and SAHFs visible as distinct DAPI-dense foci (Figure 3A). At their core, these SAHFs are enriched for heterochromatin marks, such as tri-methylation of histone H3 on lysine 9 (H3K9me3) and lysine 27 (H3K27me3), but generally exclude euchromatic markers (Figure 3A). Accordingly, chromatin layering in SAHF is associated with spatial segregation of different chromatin types and therefore demarcate active and repressed chromatin domains. Degradation of the nuclear envelope component lamin B1 and herniation of nuclear chromatin into the cytosol are well-described features of senescent cells. In accordance with this, cytosolic chromatin fragments (CCF) were visualized and quantified in 20 Gy-irradiated versus non-irradiated fibroblasts by IFM, based on their intense γH2AX staining (Figure 3B). The percentage of CCF-positive cells was significantly increased in 20 Gy-irradiated compared to non-irradiated fibroblasts (Figure 3B, Appendix A). These data suggest that radiation-induced senescence promotes the release of nuclear chromatin and the formation of CCFs.

Cyclic GMP-AMP synthase (cGAS) protein acts as cytosolic DNA sensor that binds DNA and activates the cGAS-STING interferon signaling, a critical component of the innate immune response. In the presence of nucleic acids in the cytosol cGAS catalyzes the synthesis of cGAMP, that activates the adaptor protein STING (stimulator of interferon genes), thereby triggering an innate immune type I interferon response. In order to investigate potential activation of the cGAS-STING pathway in response to cytoplasmic DNA, we measured cGAMP concentrations by a colorimetric assay (cGAMP ELISA). The concentration of cGAMP was clearly increased in 20 Gy-irradiated, but not in non-irradiated fibroblasts, suggesting that cGAS-STING signaling contributes to the production of inflammatory cytokines in senescent cells (Supplemetary Appendix A). The induction of the interferon immune response could be proven by our proteome analysis (Figure 2). 

TEM allows us to study the internal ultrastructure of nuclei and to analyze specific features of chromatin compaction with much higher spatial resolutions. Our TEM studies show that interphase chromatin of non-irradiated fibroblasts revealed homogeneous ultrastructural appearance with more electron-dense roundish structures representing the nucleoli with more compacted ribosomal DNA (Figure 3C, left panel). These nucleoli have the highest RNA synthesis rate in cell nuclei and as site of ribosome biogenesis hold a pivotal role in cell metabolism [27]. Following IR exposure, nuclei of senescent fibroblasts exhibit dramatic alterations to their higher-order chromatin structure. Chromatin compaction in spatially defined domains leads to the assembly of dense chromatin fragments (circular electron-dense bodies of 500–600 nm in diameter). These heterochromatinized fragments, sometimes closely arranged in rows, are surrounded by more decondensed areas (Figure 3C, right panel). 

### 3.4. Nuclear Grooves in Senescent Fibroblasts Visualized by TEM

High-resolution TEM was used for direct imaging the nuclear ultrastructure of senescent nuclei at nanometer resolution. While non-irradiated fibroblasts have regular nuclei without any nuclear blebs and grooves, ≈70% of irradiated fibroblasts revealed grooved nuclei with invaginations of various depths and sizes. These irregular invaginations (sometimes multiple fissures within one cell) were observed in different parts of senescent nuclei and seemed to be formed by nuclear membranes. TEM has an unparalleled resolution in X and Y dimensions, but due to limited field views and absence of Z dimensions, it can generate only fragmented 2D views of senescent cells. Therefore, we used volume electron microscopy to identify and localize deep irregular-formed invaginations in three-dimensional space. 

### 3.5. High-Resolution Imaging of Senescent Fibroblasts by SBF-SEM

Serial block-face scanning electron microscopy (SBF-SEM) is a powerful method to analyze the cellular ultrastructure in three dimensions (3D). After optimising contrasting and embedding conditions for fibroblast monolayers, the sample block was sectioned and imaged in sequence every 50 nm within the scanning electron microscope. This automated acquisition of serial-section imaging data covered the volume of entire fibroblasts (Figure 4A). The resulting EM image stacks were segmented manually for different organelles and reconstructed using AMIRA software. Figure 4 shows the 3D reconstruction of whole senescent fibroblast with accurate segmentation of the nucleus (light-blue), nucleoli (red), CCF (light-red) and lysosomes (blue) to study their characteristic morphologies (Figure 4B). 

### 3.6. SBF-SEM: Visualization of CCF Segregation

Computer-assisted segmentation and 3D reconstruction of sequential SEM images provide detailed views of the separation process of CCFs (light red) from the nucleus and their spatial relationship to lysosomes (blue) (Figure 5). Some CCFs are already spatially isolated from the main nucleus. In the cytoplasm, CCFs are targeted to the autophagy machinery, thereby initiating their lysosomal-mediated proteolytic degradation. Appendix A shows lysosomes (size ~0.5–1 µm in diameter) fusing with the outer membrane of detached CCFs (size ~1–1.2 µm in diameter), thereby initiating their autophagic degradation. 

### 3.7. SBF-SEM: Visualization of Nucleo-Cytoplasmic Nanotubes 

Our SBF-SEM datasets also allowed for the accurate segmentation of the deep invaginations towards the interior of the nucleus, which likely represent the nuclear grooves observed by 2D TEM. These tube-like structures emerge from the nuclear surface at different positions in a non-specific manner, and physically connects the cytoplasm with deeper areas of the nucleus. These tunnel-like invaginations contain inner and outer nuclear membranes. Higher-magnification micrographs clearly evidence that these invaginations touch the nucleolus and/or even cross the whole nucleus (Figure 6). The diameters of these intrusions range from 200 to 600 nm. This is very similar to the spatial dimensions obtained by our 2D TEM technique, indicating agreement and accurate quantification. Moreover, numerous electron dense particles (identical density as the nucleolus) could be visualized within these nuclear channels, suggesting a functional role for nuclear transport. 

## 4. Discussion

In the 3D nuclear space, the hierarchical genome organization into chromatin serves to precisely orchestrate cellular functions by controlling gene expression [28]. Spatial genome conformation is modulated by interactions between heterochromatin and nuclear lamina, providing peripheral tethering points to confer physiological chromatin organization. Accordingly, biophysical properties of the lamina are essential not only for the nuclear envelope stability, but also for maintaining the well-organized architecture of chromatin. Cellular senescence is characterized by the degradation of lamin B1, causing detachment of peripheral chromatin from the lamina, thereby leading to extensive chromatin restructuring. During senescence, the progressive destabilization of the nuclear envelope leads to the nucleus-to-cytoplasm blebbing of chromatin. For future studies, it will be an interesting point to study the role of CCFs in degrading nuclear components by autophagy. Previous work has shown that the autophagy protein LC3, which is involved in autophagy membrane trafficking and substrate delivery, is present in the nucleus and directly interacts with the nuclear lamina protein lamin B1, and binds to lamin-associated domains on chromatin [29]. Recent studies suggest that autophagy may be important in the regulation of cancer development and progression and in determining the response of tumor cells to anticancer therapy. The inhibition of lysosomal activities, thereby blocking the formation of autophagosomes that capture degraded components and then fuse with lysosomes to recycle these components, is a promising potential therapeutic target in cancer treatment [30].

Apart from radiation-induced senescence, cells exposed to high doses of IR can enter other modes of cell death mechanisms, such as apoptosis and necrosis [31]. Moreover, radiation-induced mitotic catastrophes may occur due to premature or improper entry of cells into mitosis in response to Dann damage and deficient cell cycle checkpoints. Disordered mitosis can produce atypical chromosome segregation and cell division and can lead to the formation of giant cells with aberrant nuclear morphology, multiple nuclei, and/or several micronuclei [31]. The generation of polyploid giant nuclei can disturb peripheral positioning of envelope-bound heterochromatin domains and can therefore disrupt the long-term balance of chromatin organization [32]. Recent work highlights the important role of lamina-associated heterochromatin domains of the structural and functional maintenance of nuclear architecture [33].

Here, we combined mass spectrometry-based proteomics and high-resolution microscopy techniques to study the hallmarks of radiation-induced senescence and to obtain insights into the pathophysiological role of nucleo-cytoplasmic chromatin blebbing. Our findings confirm the complex interrelations between chromatin restructuring in lamin-perturbed nuclei and nuclear envelope destabilization with the release of chromatin fragments into the cytosol [21]. Our results support the prevailing view that CCFs activate the cGAS-STING-dependent innate immune signaling, thereby triggering the production of SASP factors during radiation-induced senescence [20,23]. In our previous work, we showed that human WI-38 fibroblasts develop SASP with the secretion of numerous cytokines after IR exposure (analyzed at 2 weeks after 20 Gy). The pro-inflammatory cytokines interleukin-6 (IL6) and interleukin-8 (IL8), granulocyte-macrophage colony-stimulating factor (GM-CSF), and monocyte chemoattractant protein-1 (MCP1) were quantified in the supernatant of non-irradiated versus irradiated fibroblasts by enzyme-linked immunosorbent assay (ELISA). Our results showed that non-irradiated fibroblasts expressed low levels of these common SASP factors, and hence no senescence-messaging secretome. Human fibroblasts in radiation-induced senescence, by contrast, secreted high levels of IL6, IL8, GM-CSF, and MCP1, demonstrating that SASP components were significantly increased between non-senescent and senescent states in human fibroblasts [3]. Moreover, the relative gene expression of these SASP factors were quantified during radiation-induced senescence by reverse-transcriptase quantitative polymerase-chain-reaction (RT-qPCR). Gene expression analysis of WI-38 fibroblasts after senescence-inducing IR revealed significantly increased mRNA expression of IL6, CXCL8, CSF2, and CCL2 with 10–30-fold increases [3]. Collectively, these findings confirm the robust induction of SASP expression during radiation-induced senescence. Furthermore, automated image capturing and data processing of SBF-SEM provides unprecedented multi-view 3D reconstruction of the morphological organization of senescent fibroblasts. While imaging by standard diffraction limited fluorescence light microscopy shows general nuclear information with resolution in the range of ≈200 nm, our SEM data sets allow us to reconstruct 3D structures of senescent cells to the 3–5 nm resolution. The accuracy of reconstruction allows us to illustrate the complex nuclear shape of senescent cells and the precise visualization of nuclear blebs with their segregation from the main nucleus and their fusion with lysosomes. Only by high-resolution 3D imaging were we able to identify and characterize the nanoscale invaginations formed regularly in the nuclei of senescent fibroblasts. High-resolution 3D whole-cell data sets of non-irradiated fibroblasts can be obtained from the open access volume electron microscopy atlas repository [34].

Depletion of lamin B1 during senescence progression leads to segregation of A-type and B-type lamins from one another, and results in an uneven mesh layer with altered mechanical properties [14]. In some regions, lamina’s fibers begin to gap and separate, giving rise to bulges or even transient ruptures in the cell’s nuclear envelope. Accordingly, these tubular membrane structures may arise as a consequence of different mechanical forces, ‘pushing’ forces exerted, e.g., by cytoskeletal filaments, or ‘pulling’ forces by chromatin-lamin interactions and large-scale rearrangements of chromatin [14,35,36]. Another explanation of these nanoscale invaginations of the nuclear envelope may be related to the disturbed nucleo-cytoplasmic trafficking in senescent cells [37]. Senescent cells exhibit reduced responses to intrinsic and extrinsic stimuli. This diminished reaction is explained by the disrupted transmission of nuclear signals. Disruption of the nucleo-cytoplasmic trafficking is an essential feature of cellular senescence, and thus may suggest that these selective nuclear channels are involved in intracellular transport to overcome the nucleo-cytoplasmic barrier. Recent work showed that nuclear invaginations often contain cytoskeletal filaments, linked to the nuclear envelope, suggesting a direct link between regions deep inside the nuclei and cell-cell and/or cell–ECM adhesion sites [38]. Accordingly, ECM may signal through receptors, via the cytoskeleton, through nuclear matrix to chromatin to control cell- or tissue-specific function and vice versa. Potentially, nanotubes are essential 3D architectural elements of the interconnected network of the nucleus, cytoskeleton and ECM in cell communication and the structure and composition of the nuclear envelope responds to microenvironmental stimuli with important consequences for gene regulation. However, the precise pathophysiological role of these nucleo-cytoplasmic nanotubes in lamin-perturbed nuclei has to be clarified in future studies.

In the present study, the hallmark features of premature senescence triggered by ionizing radiation were analyzed in human fibroblasts, the most common cell culture system for cellular senescence. Currently, it is unclear if all cell types (even terminally differentiated cells) or only replicative or mitotically competent cells can become senescent in normal tissues [39]. Changes in nuclear fragility leading to nuclear blebbing and altered chromatin organization are indicative of a wide range of pathologies, including cancer and other age-related diseases [40]. Exploration of laminopathies, such as Hutchinson–Gilford progeria, a premature aging syndrome caused by lamin A mutations, has demonstrated the critical importance of the nuclear integrity for chromatin architecture [41]. Disruption of lamin-interacting structures have profound effects on the high-order organization of genomes and is functionally important for gene regulation. 

## 5. Conclusions

Understanding the precise pathomechanisms responsible for the formation of nuclear blebs and tubes may provide new insights into the aging process and age-related diseases. Volume electron microscopy allows for automated acquisition of serial-section imaging data that can be segmented and reconstructed, thereby providing detailed 3D views of senescent cells with precise morphologies. Unraveling complex 3D architectures is crucial for structure–function correlations to gain novel insights into the pathogenic organization of senescent cells. Our findings may help to improve the current understanding of nuclear fragility during radiation-induced senescence. 

## Figures and Tables

**Figure 1 cells-11-00273-f001:**
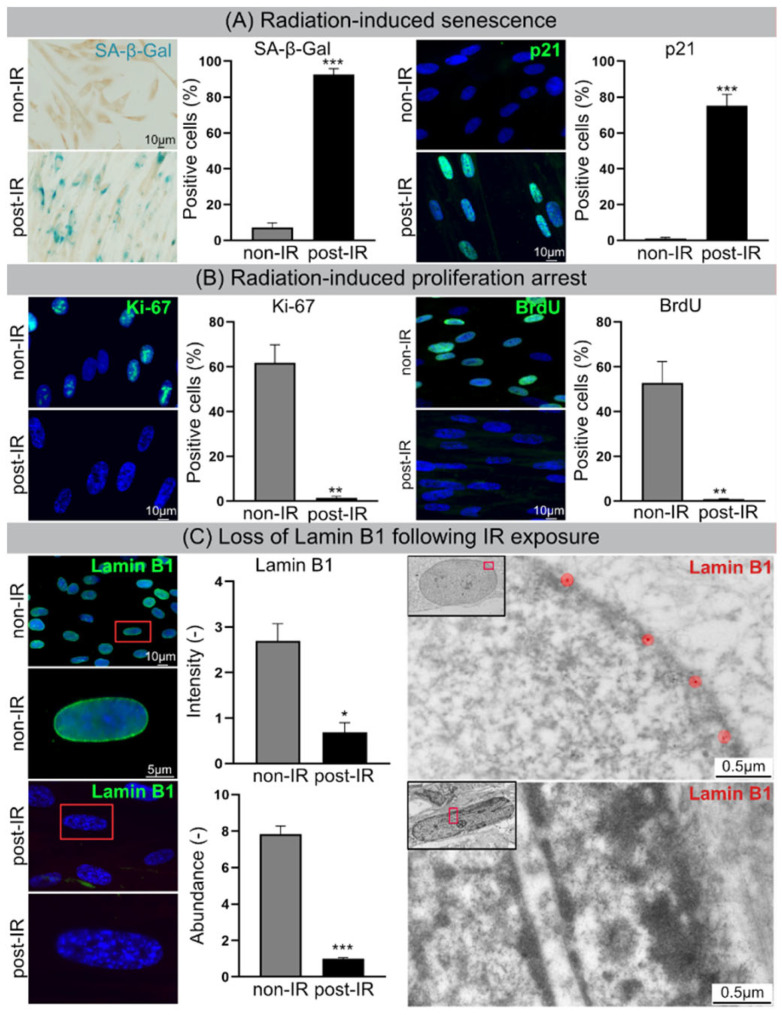
Cellular senescence following IR. (**A**) Increased numbers of SA-β-Gal-positive and p21-positive cells following IR. (**B**) Decrease in Ki-67-positive and BrdU-positive cells. (**C**) Lamin B1 loss in nuclear envelope following IR exposure visualized by IFM (left) and TEM (right). Quantification of lamin B1 in WI-38 fibroblasts by IFM (top middle), and MS (bottom middle). Data are presented as mean ± SEM, * *p* < 0.05, ** *p* < 0.01, *** *p* < 0.001.

**Figure 2 cells-11-00273-f002:**
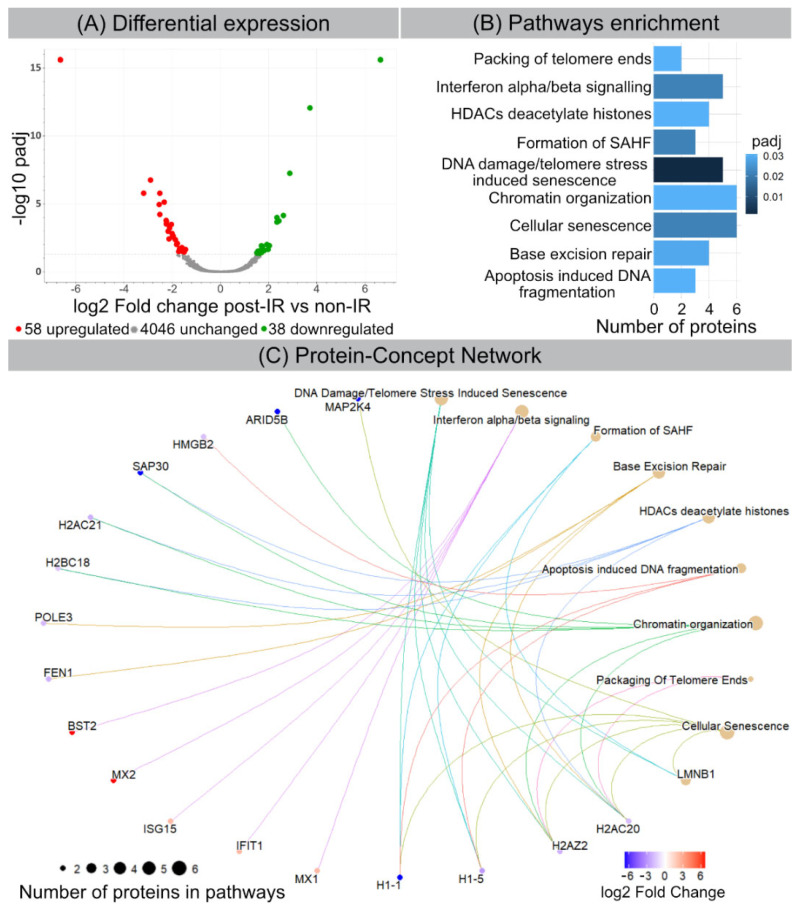
Radiation-induced changes in protein expression. (**A**) Volcano plot showing differential protein expression following IR. (**B**) Senescence-related pathway enrichment results were generated using ReactomePA R package. (**C**) Visualization of enriched pathways components by enrichplot R package.

**Figure 3 cells-11-00273-f003:**
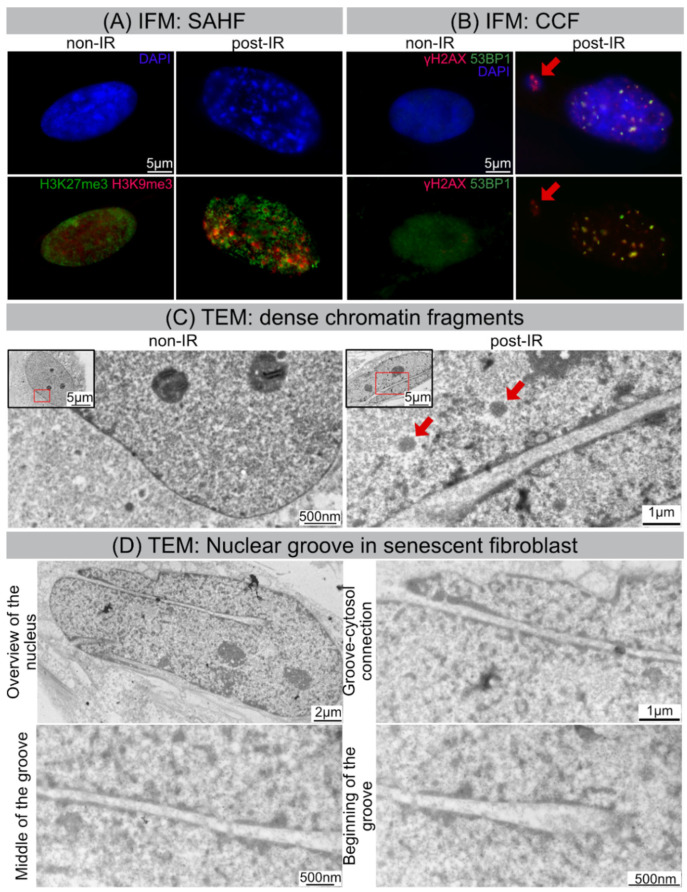
IR-induced morphological changes in WI-38 fibroblasts. (**A**) Visualization of SAHF formation by IFM. (**B**) Visualization of CCF formation by IFM. (**C**) Visualization of dense chromatin fragments (marked by red arrows) by TEM. (**D**) Visualization of nuclear groove in senescent fibroblast by TEM.

**Figure 4 cells-11-00273-f004:**
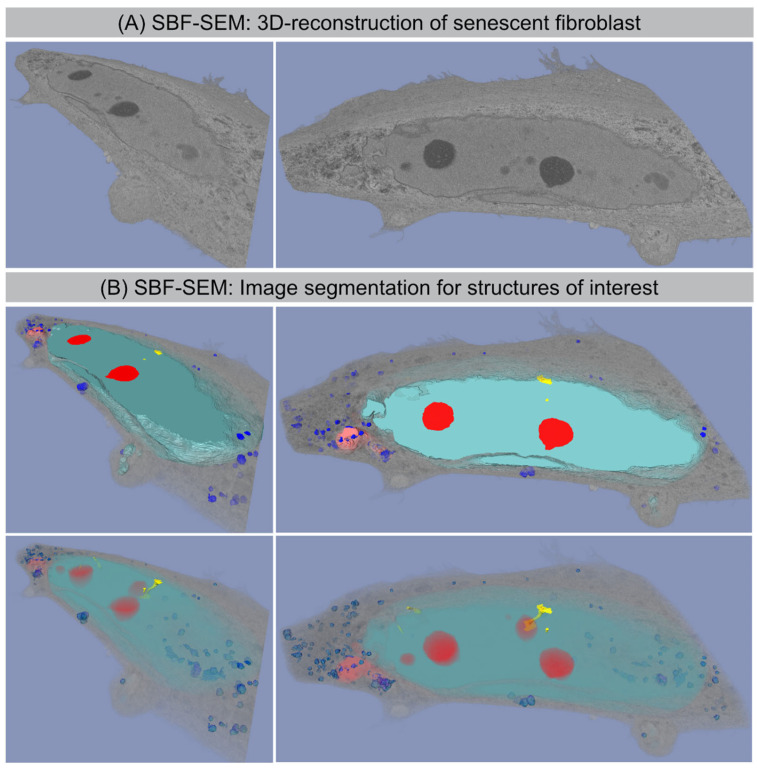
SBF-SEM: 3D reconstruction of senescent fibroblast. (**A**) Original SBF-SEM sections were used for 3D reconstruction. (**B**) Serial SBF-SEM sections were segmented for structures of interest: nucleus (light-blue), nucleoli (red), lysosomes (blue), and CCF (light-red), nanotubes (yellow), cytosol (gray) and used for 3D visualization (bottom row: transparent view).

**Figure 5 cells-11-00273-f005:**
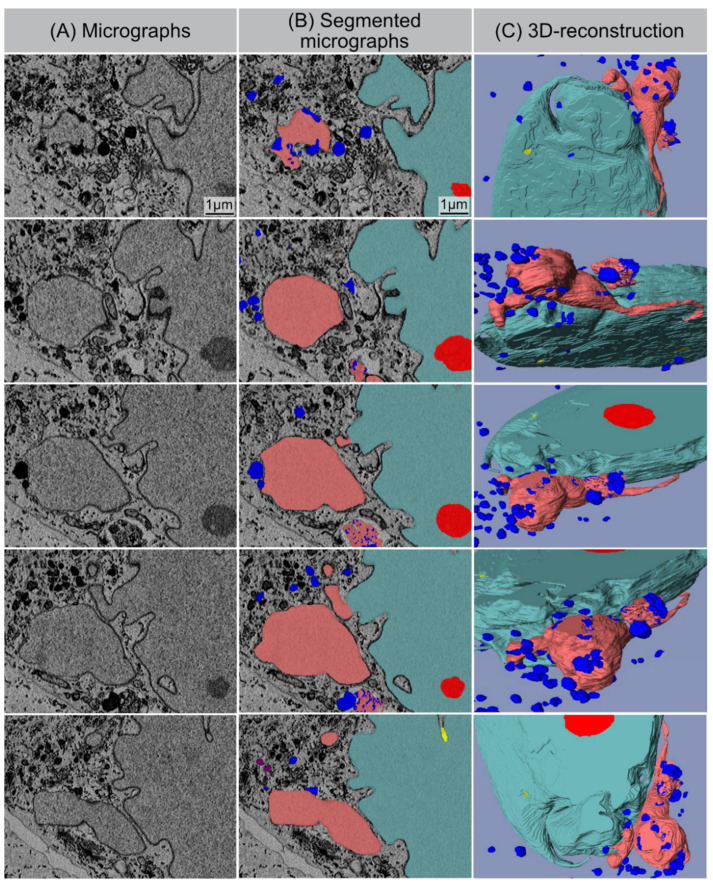
SBF-SEM: Separation process of CCF. (**A**) Original micrographs presenting detaching CCF. (**B**) Segmented micrographs of the same regions. (**C**) 3D reconstruction of detaching CCF: nucleus (light-blue), nucleoli (red), lysosomes (blue), and CCF (light-red), nanotubes (yellow).

**Figure 6 cells-11-00273-f006:**
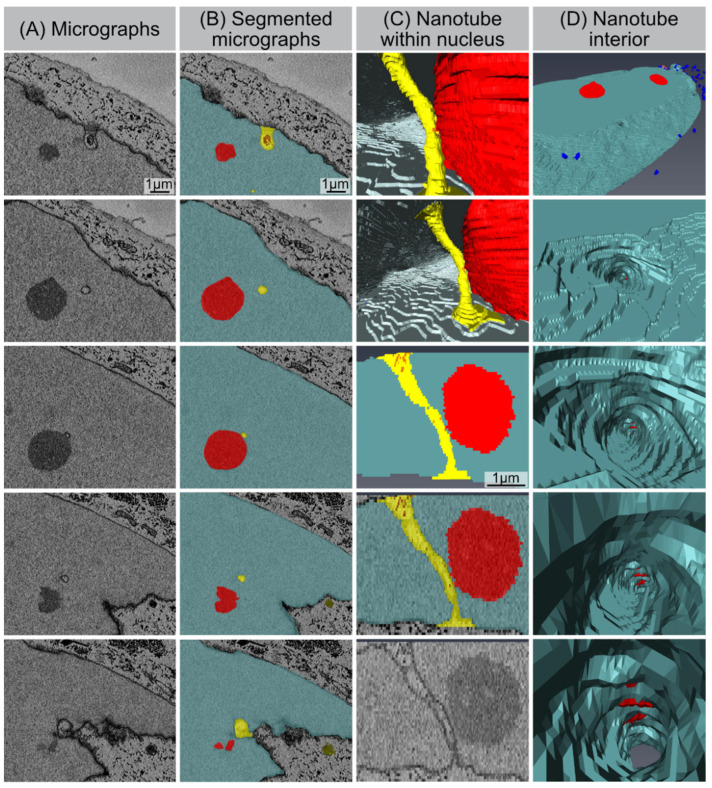
Nucleo-cytoplasmic nanotube. (**A**) Original micrographs showing cross-sections of the nanotube. (**B**) Segmented micrographs for the same area: nucleus (light-blue), nucleoli (red), nanotubes (yellow). (**C**) Models showing nanotube’s appearance from within the nucleus (1st and 2nd image from top), segmented volume showing the nanotube from side (3rd and 4th image from top), and original micrograph of the same region. (**D**) Visualization of the nanotube’s interior.

## Data Availability

The data that support the findings of this study are available from the Department of Radiation Oncology, Saarland University Hospital, Homburg/Saar, Germany.

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
