# Peer review of "Nuclear Fragility in Radiation-Induced Senescence: Blebs and Tubes Visualized by 3D Electron Microscopy"

_cells, 2022, doi:10.3390/cells11020273_

Round 1

Reviewer 1 Report

The results shown in the manuscript are based only on human WI-38 fibroblasts. Hence, data obtained may be considered preliminary needing additional assays in diverse cell types in order to corroborate the conclusions. Moreover, it is not clear and explained why the Authors used fibroblasts as model system in the study given that these cells exhibit peculiar features respect to other cells from different tissues. In this respect,  data should be corroborated adequately to make suitable conclusions.

Author Response

Line 51-54: „As cellular senescence is characterized as the basis for aging in organisms, culturing of normal human fibroblasts to mimic the in-vivo aging processes has been developed as major methods to investigate cellular and molecular events involved in aging (Chen H et al. 2013).“

Line 364-367: „In the present study, the hallmark features of premature senescence triggered by ionizing radiation were analyzed in human fibroblasts, the most common cell culture system for cellular senescence. Currently, it is unclear if all cell types (even terminally differentiated cells) or only replicative or mitotically competent cells can become senescent in normal tissues (von Zglinicki et al., 2021).

However, based on our findings, the specific features of radiation-induced senescence can be evaluated for all different cells types, preferentially in complex co-cultivation systems.

Reviewer 2 Report

In this accurate works Freyter and collegues shown how irradiation ionizing radiation (IR) triggers prolonged DNA damage response and induces premature senescence activating canonical markers in human fibroblasts. Throug electron microscopy, authors identified nanotubular channels formed in lamin-perturbed nuclei of senescent fibroblasts with a potential role for expulsion and degradation of degraded chromatin.

The paper is clear and well written, however some minor or major flaws need to be elucidated, in details:

Major points:

  • Protocol for Cytochemical detection of SA-β-Gal is different from typical used ones, as seems to involve use of secondary antibody instead of classical colorimetric, but the result in figure 1 seems to be colorimetric. Authors should clarify.
  • In results 3.2 authors state that “interferon signaling may drive the production of SASP”. Is it possible to measure if there is an actual increase of released sasp and which one are the most involved?
  • Are CCF positive for markers possibly involved in their degradation as LC3?
  • Do the formation of nanotubules occur in every irradiated cell or in a fraction? Is it present in any non-irradiated cells? Is it possible in this case to quantify the induction of this structure formation?
  • A Representative SBS-SEM images of control cell should be provided at least as supplementary, to allow a better comparison with irradiated cells (fig 4b)
  • It would be interesting to understand if the blockage of lysosomial activity by inhibitors would block also fusion of CCF with lysosomes, is it possible visualize it with high-resolution 3D imaging? Could it provoke an accumulation of CCF in the cells and analyze the effects?
  • Nanotubles discussion should be extended

Minor points:

  • There are some acronyms in the introduction and methods sections that are not explained.
  • In introduction few words about cGAS-STING-dependent innate immune signaling would help the reader
  • Antibody list should include clone name or antibody catalog number.
  • In results 3.1 authors should better explain how they meseaure Laminin B1 levels, what are Abundance (-) and Intencity (-) reported in figure 1C exactly? Is stated that Laminin and IFNN are meseaured by proteomics but not if the data are reported in supp material.
  • In results 3.2 is stated “Altered proteins in affected pathways are listed in supplementary table 4”, that is a bit misleading, as in table 4 are reported only the pathways regulated but not the single protein.
  • Figure 6 lacks colours legend
  • Some figure legends should be provided for supplementary material

Author Response

Major points:

  • Protocol for Cytochemical detection of SA-β-Gal is different from typical used ones, as seems to involve use of secondary antibody instead of classical colorimetric, but the result in figure 1 seems to be colorimetric. Authors should clarify.

We would kindly like to apologize for the misleading description. In addition to the colorimetric detection of SA-β-Gal we stained for the histone variant H2A.J to quantify senescent fibroblasts. The results for H2A.J have been published previously (Isermann A, et al. 2020).

Line 108-117: “Following 5-min fixation with 2% paraformaldehyde and 0.2% glutaraldehyde, cells were incubated with X-Gal staining solution (AppliChem GmbH, Darmstadt, Germany) at 37 °C overnight. After 30s methanol incubation, dried samples were permeabilized with 0.2% Triton X-100 and washed with 1% BSA. Samples were blocked with 4% BSA for 1h, followed by an overnight incubation with anti-H2A.J primary antibody. Incubation with Dako immunoglobulin/bioatinylated secondary antibody (Agilent, Waldbronn, Germany) was followed by Vectastain ABC Peroxidase standard (Vector, Burlingame, CA, USA) and SIGMAFAST™ 3.3′ Diaminobenzidine (Merck, Darmstadt, Germany) incubations, respectively. Samples were finally mounted in Dako Faramount Mounting Medium (Agilent, Waldbronn, Germany).”

  • In results 3.2 authors state that “interferon signaling may drive the production of SASP”. Is it possible to measure if there is an actual increase of released sasp and which one are the most involved?

Compared the new papagraph (line 412-428):

In our previous work we have already shown that human WI-38 fibroblasts develop SASP with the secretion of numerous cytokines after IR exposure (at 2 weeks after 20Gy). The pro-inflammatory cytokines interleukin-6 (IL6) and interleukin-8 (IL8), granulocyte-macrophage colony-stimulating factor (GM-CSF), and monocyte chemoattractant protein-1 (MCP1) were quantified in the supernatant of non-irradiated versus irradiated fibroblasts by enzyme-linked immunosorbent assay (ELISA). Our results showed that non-irradiated fibroblasts expressed low levels of these common SASP factors, and hence no senescence-messaging secretome. Human fibroblasts in radiation-induced senescence, by contrast, secreted high levels of IL6, IL8, GM-CSF, and MCP1, demonstrating that SASP components were significantly increased between non-senescent and senescent states in human fibroblasts (Isermann et al. 2020). Moreover, the relative gene expression of these SASP factors were quantified during radiation-induced senescence by reverse-transcriptase quantitative polymerase-chain-reaction (RT-qPCR). Gene expression analysis of WI-38 fibroblasts after senescence-inducing IR revealed significantly increased mRNA expression of IL6, CXCL8, CSF2, and CCL2 with 10–30-fold increases (Isermann et al. 2020). Collectively, these findings confirm the robust induction of SASP expression during radiation-induced senescence.

  • Are CCF positive for markers possibly involved in their degradation as LC3?

Until now we didn´t check the role of CCFs in degrading nuclear components by autophagy. For future studies it will be an interesting point to analyze the autophagy protein LC3 which is involved in autophagy membrane trafficking and substrate delivery.

Line 383-388: „For future studies it will be an interesting point to study the role of CCFs in degrading nuclear components by autophagy. Previous work has shown that the autophagy protein LC3, which is involved in autophagy membrane trafficking and substrate delivery, is present in the nucleus and directly interacts with the nuclear lamina protein lamin B1, and binds to lamin-associated domains on chromatin (Dou Z. et al. 2015).“

  • Do the formation of nanotubules occur in every irradiated cell or in a fraction? Is it present in any non-irradiated cells? Is it possible in this case to quantify the induction of this structure formation?

Compare line 316-318: “While non-irradiated fibroblasts have regular nuclei without any nuclear blebs and grooves, ≈70% of irradiated fibroblasts revealed grooved nuclei with invaginations of various depths and sizes.

  • A Representative SBS-SEM images of control cell should be provided at least as supplementary, to allow a better comparison with irradiated cells (fig 4b)

Due to the great technical effort SBF-SEM was performed only for senescent fibroblasts after IR exposure. Normal non-irradiated fibroblasts were analyzed by our TEM approach.

Compare line 437-439: „High-resolution 3D whole-cell data sets of non-irradiated fibroblasts can be obtained from the open access volume electron microscopy atlas repository (c.Shan Sung et al. 2021).

  • It would be interesting to understand if the blockage of lysosomial activity by inhibitors would block also fusion of CCF with lysosomes, is it possible visualize it with high-resolution 3D imaging? Could it provoke an accumulation of CCF in the cells and analyze the effects?

The inhibition of lysosomal activities may play a critical role in autophagy and may emerge as an attractive target in cancer therapy. However, we didn´t analyze the blockage of lysosomial activity by inhibitors, thereby potentially blocking the fusion of CCF with lysosomes.

Compare new paragraph in line 388-393: „Recent studies suggest that autophagy may be important in the regulation of cancer development and progression and in determining the response of tumor cells to anticancer therapy. The inhibition of lysosomal activities, thereby blocking the formation of autophagosomes that capture degraded components and then fuse with lysosomes to recycle these components, is a promising potential therapeutic target in cancer treatment.

  • Nanotubles discussion should be extended

Line 453-462: “Recent work has shown that nuclear invaginations often contain cytoskeletal filaments, linked to the nuclear envelope, suggesting a direct link between regions deep inside the nuclei and cell–cell and/or cell–ECM adhesion sites (Jorgens DM 2017). Accordingly, ECM may signal through receptors, via the cytoskeleton, through nuclear matrix to chromatin to control cell- or tissue-specific function and vice versa. Potentially, nanotubes are essential 3D architectural elements of the interconnected network of the nucleus, cytoskeleton and ECM in cell communication and the structure and composition of the nuclear envelope responds to microenvironmental stimuli with important consequences for gene regulation.”

Minor points:

  • There are some acronyms in the introduction and methods sections that are not explained.

cGAS: cyclic GMP-AMP synthase; STING: stimulator of interferon genes; ATCC (American Type Culture Collection); MEM (Minimum Essential Media; ThermoFisher Scientific); PBS (phosphate-buffered solution; ThermoFisher Scientific); BSA (bovine serum albumin); RIPA (radio-immunoprecipitation assay) buffer

  • In introduction few words about cGAS-STING-dependent innate immune signaling would help the reader

New paragraph in line 82-88: „This cGAS-cGAMP-STING signaling pathway is a first line defence component of the innate immune system that mediates type I interferon inflammatory responses to repel viral and bacterial infections. Only recently it has been shown that the cGAS-STING signaling is also activated by endogenous DNA aberrantly localized in the cytosol, such as extranuclear chromatin resulting from genotoxic stress (Hopfner KP 2020).“  

  • Antibody list should include clone name or antibody catalog number.

The list of antibodies is now included as supplementary table 5.

  • In results 3.1 authors should better explain how they meseaure Laminin B1 levels, what are Abundance (-) and Intensity (-) reported in figure 1C exactly? Is stated that Laminin and IFNN are meseaured by proteomics but not if the data are reported in supp material.

Relative fluorescence intensity of lamin B1 was measured based on the FITC signal (normalized to DAPI) in non-irradiated versus irradiated fibroblasts using Nikon NIS-Elements Basic Research acquisition software. Relative protein abundance of lamin B1 in non-irradiated versus irradiated fibroblasts was measured using HPLC-MS/MS for the proteome analysis (LMNB1 in supplementary table 1).

Line 226-230: “For lamin B1 the relative fluorescence intensity (IFM: FITC signal normalized to DAPI,Nikon NIS-Elements) and relative protein abundance (proteome analysis by HPLC-MS/MS: LMNB1 in supplementary table 1) was measured in non-irradiated versus irradiated fibroblasts.

  • In results 3.2 is stated “Altered proteins in affected pathways are listed in supplementary table 4”, that is a bit misleading, as in table 4 are reported only the pathways regulated but not the single protein.

We agree that this statement is misleading; therefore we refer to supplementary table 4 after the sentence, Line 259: „Accordingly, our proteome data are in line with previous studies, showing that cellular senescence is characterized by extensive chromatin restructuring with global histone H1 loss and SAHF formation, as well as with nuclear degradation of lamin B1 (supplementary table 4).

  • Figure 6 lacks colours legend

Line 368-369: “(B) Segmented micrographs for the same area: nucleus (light-blue), nucleoli (red), nanotubes (yellow).“

  • Some figure legends should be provided for supplementary material

Line 483-491:

„Supplementary figure 1: Principal component analysis (PCA): Pattern analysis of the investigated data sets for irradiated (20Gy post-IR) and non-irradiated (non-IR) fibroblasts.

Supplementary figure 2: CCF formation and cGAMP concentration:

Quantification of CCF-positive cells by IFM for non-irradiated (non-IR) and irradiated (20Gy, 2 weeks post-IR) fibroblasts (right panel)

Quantification of cGAMP concentration: Quantification of cGAMP protein relative to total protein (g= gram) in non-irradiated (non-IR) and irradiated (20Gy, 2 weeks post-IR) fibroblasts (left panel).

Supplementary figure 3: SBF-SEM: fusion of CCF with lysosomes

High-resolution display of the fusion of CCF with lysosomes to become autolysosomes, in which the sequestered cargos are degraded and recycled for the maintenance of cellular homeostasis.“

Reviewer 3 Report

The article “Nuclear fragility in radiation-induced senescence: Blebs and tubes visualized by 3D electron microscopy” by Freyter et al. addresses novel electron microscopy approaches in combination with mass-spectrometry based proteomics for investigations of pathophysiology of the senescence-response.  By novel serial block-face scanning electron microscopy obtaining whole cell datasets the morphological organization of senescent fibroblasts was investigated, disruption of nuclear lamin organization leading to extensive chromatin restructuring was observed, and destabilization of the nuclear membrane with release of chromatin fragments was visualized.

The article is well written and demonstrates the potentials of the methodological approaches for further systematic investigations towards understanding of the mechanistic connections for radiation-induced DNA damage causing senescence.

Some minor point should be considered before the article could be accepted:

1.) line 292: “(Fig. 3C….” not 2c

2.) line 352/353: A new publication addressing interactions of chromatin organization and lamina can be considered:
https://doi.org/10.3390/cells10071582

3.)line 371: here you refer to fluorescence microscopy but the resolution limit is true for standard diffraction limited fluorescence light microscopy. Super-resolution light/fluorescence microscopy techniques have a much better resolution. It should be corrected.

4.) In the discussion it may be mentioned that for heavily treated (20 Gy) cells, senescence is not the only fate. They can run into apoptosis or even survive with long term chromatin re-organization (see for instance: 
http://dx.doi.org/10.1016/j.mrgentox.2013.05.004

Author Response

Reviewer #3

The article “Nuclear fragility in radiation-induced senescence: Blebs and tubes visualized by 3D electron microscopy” by Freyter et al. addresses novel electron microscopy approaches in combination with mass-spectrometry based proteomics for investigations of pathophysiology of the senescence-response.  By novel serial block-face scanning electron microscopy obtaining whole cell datasets the morphological organization of senescent fibroblasts was investigated, disruption of nuclear lamin organization leading to extensive chromatin restructuring was observed, and destabilization of the nuclear membrane with release of chromatin fragments was visualized.

The article is well written and demonstrates the potentials of the methodological approaches for further systematic investigations towards understanding of the mechanistic connections for radiation-induced DNA damage causing senescence.

Some minor point should be considered before the article could be accepted:

1.) line 292: “(Fig. 3C….” not 2c

Line 313: The description in the text refers to Fig. 3C.

2.) line 352/353: A new publication addressing interactions of chromatin organization and lamina can be considered:
https://doi.org/10.3390/cells10071582

Line 402-404: „Accordingly, recent publications highlight the important role of lamina-associated heterochromatin domains for the structural and functional maintenance of nuclear organization (Erenepreisa J 2021).“

3.)line 371: here you refer to fluorescence microscopy but the resolution limit is true for standard diffraction limited fluorescence light microscopy. Super-resolution light/fluorescence microscopy techniques have a much better resolution. It should be corrected.

Line 431: „While imaging by standard diffraction limited fluorescence light microscopy …

4.) In the discussion it may be mentioned that for heavily treated (20 Gy) cells, senescence is not the only fate. They can run into apoptosis or even survive with long term chromatin re-organization (see for instance: 
http://dx.doi.org/10.1016/j.mrgentox.2013.05.004

Line 394- 404: „Apart from radiation-induced senescence, cells exposed to high doses of IR can enter other modes of cell death mechanisms, such as apoptosis and necrosis (Eriksson D et al. 2010). Moreover, radiation-induced mitotic catastrophes may occur due to premature or improper entry of cells into mitosis in response to DNA damage and deficient cell cycle checkpoints. Disordered mitosis can produce atypical chromosome segregation and cell division and can lead to the formation of giant cells with aberrant nuclear morphology, multiple nuclei, and/or several micronuclei (Eriksson D et al. 2010). The generation of polyploid giant nuclei can disturb peripheral positioning of envelope-bound heterochromatin domains and can therefore disrupt the long-term balance of chromatin organization (Schwarz-Finsterle J. et al. 2013 et al.). Recent work highlights the important role of lamina-associated heterochromatin domains for the structural and functional maintenance of nuclear organization (Erenepreisa J 2021).“

Round 2

Reviewer 1 Report

The revised version of the manuscript did not address the comments of the Reviewer.

Reviewer 2 Report

Authors have improved the text clarifying the context of some sentences in the introduction, and fixing some issue in references/legends/figures. Although some mechanisms are not fully described, the paper worth the publication in the present form due to the novelty introduced and the technical accuracy.